# Evaluation of both exonic and intronic variants for effects on RNA splicing allows for accurate assessment of the effectiveness of precision therapies

Anya T. Joynt[1], Taylor A. Evans[1], Matthew J. Pellicore[1], Emily F. Davis-Marcisak[1], Melis A. Aksit[1], Alice C. Eastman[1], Shivani U. Patel[2], Kathleen C. Paul[1], Derek L. Osorio[1], Alyssa D. Bowling[1], Calvin U. Cotton[3], Karen S. Raraigh[1], Natalie E. West[2], Christian A. Merlo[2], Garry R. Cutting[1]*, Neeraj Sharma[1]*

1 McKusick-Nathans Institute of Genetic Medicine, Johns Hopkins University School of Medicine, Baltimore, Maryland, United States of America, 2 Division of Pulmonary and Critical Care Medicine, Department of Medicine, Johns Hopkins Hospital, Baltimore, Maryland, United States of America, 3 Departments of Pediatrics, Physiology and Biophysics, Case Western Reserve University, Cleveland, Ohio, United States of America

* gcutting@jhmi.edu (GRC); nsharma5@jhmi.edu (NS)

**Data Availability Statement:** All relevant data are within the manuscript and its Supporting Information files.

## Abstract

Elucidating the functional consequence of molecular defects underlying genetic diseases enables appropriate design of therapeutic options. Treatment of cystic fibrosis (CF) is an exemplar of this paradigm as the development of CFTR modulator therapies has allowed for targeted and effective treatment of individuals harboring specific genetic variants. However, the mechanism of these drugs limits effectiveness to particular classes of variants that allow production of CFTR protein. Thus, assessment of the molecular mechanism of individual variants is imperative for proper assignment of these precision therapies. This is particularly important when considering variants that affect pre-mRNA splicing, thus limiting success of the existing protein-targeted therapies. Variants affecting splicing can occur throughout exons and introns and the complexity of the process of splicing lends itself to a variety of outcomes, both at the RNA and protein levels, further complicating assessment of disease liability and modulator response. To investigate the scope of this challenge, we evaluated splicing and downstream effects of 52 naturally occurring *CFTR* variants (exonic = 15, intronic = 37). Expression of constructs containing select *CFTR* intronic sequences and complete *CFTR* exonic sequences in cell line models allowed for assessment of RNA and protein-level effects on an allele by allele basis. Characterization of primary nasal epithelial cells obtained from individuals harboring splice variants corroborated *in vitro* data. Notably, we identified exonic variants that result in complete missplicing and thus a lack of modulator response (e.g. c.2908G>A, c.523A>G), as well as intronic variants that respond to modulators due to the presence of residual normally spliced transcript (e.g. c.4242+2T>C, c.3717+40A>G). Overall, our data reveals diverse molecular outcomes amongst both exonic and intronic variants emphasizing the need to delineate RNA, protein, and functional effects of each variant in order to accurately assign precision therapies.

**Funding:** This work was supported by the following grants: CF Foundation (Sharma 19I0), and CF Research Scholar Program (Gilead Sciences) to NS; R01DK44003 to GRC; CF Foundation Cotton14XX0, and R44HL134012 to CUC. NS was awarded the Vertex Research Innovation Award outside the submitted work. The funders had no role in study design, data collection and analysis, decision to publish, or preparation of the manuscript.

**Competing interests:** The authors have declared that no competing interests exist.

## Author summary

Genetic variants that impact pre-mRNA splicing are a common cause of genetic disease and have varying downstream molecular consequences. As a result, precision therapies that function at the protein level are not always effective for these variants and thus careful assessment is necessary. Here we evaluate RNA-level effects of 52 variants in the cystic fibrosis transmembrane conductance regulator (*CFTR*) gene and show that study of splicing and its consequences allows for more accurate assignment of precision therapies.

## Introduction

Splicing of the pre-mRNA to produce mature transcript is a complex process that depends on specific sequence motifs [1–6]. Thus, it is not surprising that genetic variants can disrupt this process through a variety of mechanisms including weakening canonical splice sites, activation of cryptic splice sites, and alteration of splice regulatory sequences. While some of this variation is tolerated and preserves normal splicing to varying degrees (e.g. GC donor splice sites [6–9]), many of these variants completely preclude production of full-length protein. In fact, genetic variants that impact splicing have been estimated to account for anywhere from 10% to 50% of disease causing variants [10–12]. The disease liability of variants affecting the canonical splice sites (5'GT and 3'AG) has been long understood since early identification in β-thalassemia and phenylketonuria [7, 13, 14]. While variants that act on splicing in different positions are more varied in their consequences, they are well documented causes of a variety of inherited conditions including β-thalassemia [13], Duchenne muscular dystrophy (DMD) [15], Hutchinson-Gilford progeria syndrome (HGPS) [16], and cystic fibrosis (CF) [17–20]. CF is an autosomal recessive, multisystem disorder affecting approximately 70,000 individuals worldwide and is caused by variants in the cystic fibrosis transmembrane conductance regulator (*CFTR*) gene, which encodes an epithelial chloride channel [21]. Key phenotypic features include elevated sweat chloride (>60mmol/L), diminished lung function, high risk of respiratory infections, and pancreatic insufficiency [21]. Allelic heterogeneity makes CF an informative model for the study of genetic disease as ~2,000 variants (CFTR Mutation Database) have been identified in the *CFTR* gene, ~10% of which are thought to impact splicing (CFTR Mutation Database). Notably, the severity of clinical presentations of CF have been well correlated with the deleteriousness of *CFTR* variants [22]. Identifying the disease liability of variants and their specific mechanism of disease has become critical as molecular therapies for CF target the underlying molecular defect, thus applying to select individuals depending on the genetic variants they harbor [21].

As these treatments expand to include more and more individuals with variants affecting protein processing and function, understanding which variants allow for production of protein becomes imperative. While these drugs, known as CFTR modulators, were initially developed with specific and generally more common variants in mind (i.e. F508del, G551D), label expansion has since allowed for treatment of additional *CFTR* variants [23, 24]. Notably, *in vitro* data has been used in support of these expansions when *in vivo* data cannot be obtained [25]. The recent development of a highly effective triple-combination therapy (Trikafta) has expanded treatment to include 90% of individuals with CF [26]. However, the remaining 10% include individuals with rare exonic and intronic variants that have yet to be thoroughly characterized. The drug response of these variants is unknown and unlikely to be evaluated in a clinical trial setting. Therefore, it is paramount that *in vitro* data accurately reflects *in vivo*

circumstances to allow for expansion of these therapies as well as determination of which individuals will require alternative treatment options.

Typical *in vitro* assessment of *CFTR* variants has focused on cDNA based systems allowing only protein processing and function to be evaluated [27–30]. While some variants have been considered for RNA-level effects, hybrid minigene systems are often employed, which only allow for assessment of splicing [31–33]. We and others have previously established expression minigenes (EMGs) as a useful model for studying RNA and protein simultaneously [34–39]. Assessment of the downstream consequences of missplicing allows for precise determination of which variants may be eligible for modulator treatment.

While 'missense' variants are often assumed to cause disease through an impact at the protein-level, and thus assumed to respond to modulators, intronic variants can be incorrectly assumed to allow for no protein production at all and thus no drug response. Here we consider both of these mechanisms and focus on evaluating individual nucleotide changes from RNA-level impacts to ultimate functional consequences in order to assess therapeutic options. By studying individual variants in heterologous expression systems, we determine molecular mechanism and drug response on an allele by allele basis. Complementary assessment in primary nasal epithelial cells derived from individuals with CF allows for confirmation of our *in vitro* data.

## Results

### Predicted effects of exonic variants overlook the impact of single nucleotide changes at the RNA level

Exonic variants resulting in a single nucleotide change that could produce an amino acid substitution are typically assumed to be missense variants and are often referred to by their predicted effect at the protein level [40, 41], rather than by their HGVS designation (which we use here for clarity) [42]. This system of evaluating variants overlooks the potential for these nucleotide changes to have RNA-level effects. Here we evaluate exonic *CFTR* variants for effects on mRNA splicing. Given that the consensus splice sites extend into the exons, we chose naturally occurring *CFTR* variants located in the first and second (n = 6 beginning of exon variants) as well as penultimate and last (n = 5 end of exon variants) nucleotide of an exon. Additionally, we evaluated the splice-defect potential of *CFTR* exonic variants that were predicted by the splice algorithm CryptSplice [37], but not located at an intron-exon junction (n = 4 'middle' of exon variants). In total, we chose 15 exonic variants for study, all of which were subsequently evaluated by both CryptSplice and an additional splice prediction tool (SpliceAI, [43]). One variant was assessed in primary human nasal epithelial (HNE) cells and 14 were introduced into expression minigenes (EMGs). EMGs are plasmids containing all of the exons of *CFTR* and select full-length or abridged *CFTR* intronic sequences (**Fig 1**). These constructs allow for evaluation of the effects of *CFTR* variants on mRNA splicing of adjacent introns and have been previously shown to faithfully recapitulate *in vivo* mRNA splicing [35, 36, 38].

After verifying that the EMGs used in this study generate normally spliced RNA, we used site-directed mutagenesis to introduce 14 exonic variants into relevant EMGs and evaluated their effect on splicing by transfecting HEK293 cells. RT-PCR and Sanger sequencing revealed normal splicing of six variants, while eight variants resulted in production of misspliced *CFTR* mRNA isoforms (**Table 1**). Of these eight variants, one was located in the first nucleotide of an exon (c.274G>A), three were located in the middle (outside the consensus splice site) of an exon (c.454A>G, c.523A>G, c.2816A>G), and four were located in the last nucleotide of an exon (c.2908G>C, c.2908G>A, c.3717G>C, c.3873G>C). We did not find any variants in the second or penultimate nucleotide of an exon that resulted in misspliced products (c.581G>T,

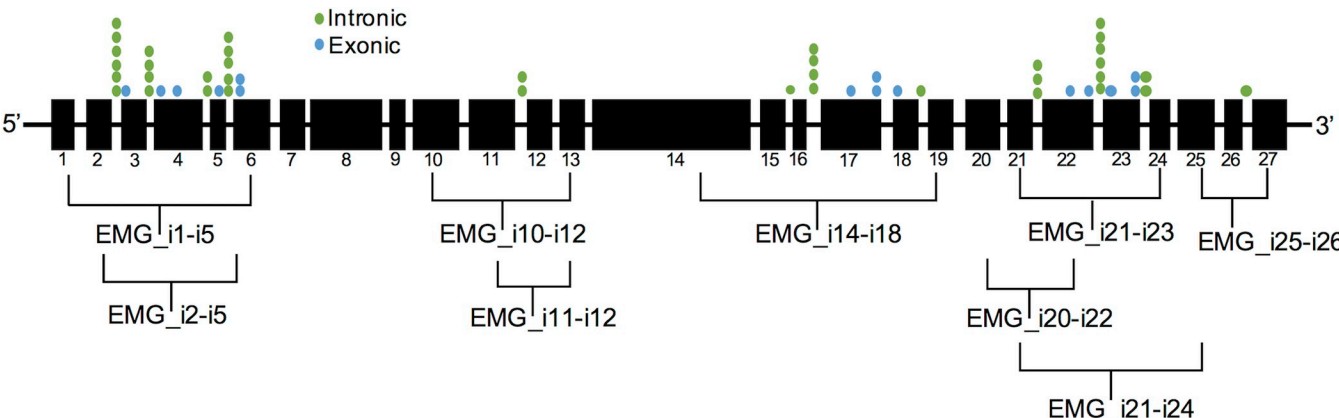

**Fig 1. Graphical representation of *CFTR* gene and location of variants analyzed.** Black boxes indicate exons and their size relative to other exons; lines between represent introns that are not to scale. Brackets indicate regions encompassed by expression minigenes (EMG, i: intron). Dots show location of variants studied (green: intronic, blue: exonic).

c.3719T>G or c.3872A>G, c.166G>A). Overall, two of the eight misspliced variants produced both misspliced and normally spliced transcript (c.274G>A, c.3873G>C), allowing for production of the full-length protein isoform that would be predicted by looking at the nucleotide

**Table 1. Exonic variants evaluated for impact on *CFTR* mRNA splicing.**

| BEGINNING OF EXON (FIRST AND SECOND NUCLEOTIDE) | | | | | |
|---|---|---|---|---|---|
| **HGVS** | **Predicted Effects** | | **Experimental Outcomes** | | |
| Nucleotide change | Predicted effect (Legacy) | Splicing (*in silico* prediction)* | RNA effect | Protein present | Modulator response |
| c.166 G>A | p.Glu56Lys (E56K) | Does not missplice | Does not missplice | p.Glu56Lys | Responsive[†] |
| c.274 G>A | p.Glu92Lys (E92K) | Does not missplice | Missplices[‡] | p.Glu92Lys | Responsive[78] |
| c.580 G>A | p.Gly194Arg (G194R) | Does not missplice | Does not missplice | p.Gly194Arg | — |
| c.581 G>T | p.Gly194Val (G194V) | Does not missplice | Does not missplice | p.Gly194Val | — |
| c.2909 G>A | p.Gly970Asp (G970D)[46] | Indeterminate | Does not missplice | p.Gly970Asp | Responsive |
| c.3719 T>G | p.Val1240Gly (V1240G) | Does not missplice | Does not missplice | p.Val1240Gly | — |
| MIDDLE OF EXON (OUTSIDE OF SPLICE SITE) | | | | | |
| Nucleotide change | Predicted effect (Legacy) | Splicing (*in silico* prediction)* | RNA effect | Protein present | Modulator response |
| c.454 A>G | p.Met152Val (M152V)[37] | Missplices | Missplices | shortened | Non-responsive |
| c.523 A>G | p.Ile175Val (I175V) | Missplices | Missplices | shortened | — |
| c.2816 A>G | p.His939Arg (H939R)[37] | Missplices | Missplices | shortened | — |
| c.3700 A>G | p.Ile124Val (I1234V)[44] | Missplices | Missplices | p.Ile1234Val | Responsive |
| END OF EXON (PENULTIMATE AND LAST NUCLEOTIDE) | | | | | |
| Nucleotide change | Predicted effect (Legacy) | Splicing (*in silico* prediction)* | RNA effect | Protein present | Modulator response |
| c.2908 G>C | p.Gly970Arg (G970R)[46] | Missplices | Missplices | shortened | Non-responsive[45] |
| c.2908 G>A | p.Gly970Ser (G970S) | Missplices | Missplices | shortened | — |
| c.3717 G>C | p.Arg1239Ser (R1239S) | Missplices | Missplices | shortened | — |
| c.3872 A>G | p.Gln1291Arg (Q1291R) | Does not missplice | Does not missplice | p.Gln1291Arg | — |
| c.3873 G>C | p.Gln1291His (Q1291H)[50] | Indeterminate | Missplices | p.Gln1291His | Responsive |

The numbers shown in the superscript indicate previously published study on the respective variant.

Data supporting these findings in **S1 Data**.

*A prediction was deemed "indeterminate" if the two *in silico* tools used did not agree (**S1 Table**).

[†]This variant is approved for modulator therapy.

[‡]The primary RNA isoform produced by this variant is normally spliced (**S1 Fig**).

substitution as a missense variant (**Table 1**). While misspliced transcript was the major isoform observed for c.3873G>C (studied in detail below), c.274G>A favored the normally spliced product (**S1 Fig**). Additionally, we studied one exonic variant known to missplice (c.3700A>G, [44]) in HNE cells collected from an individual homozygous for this variant and identified the same misspliced isoform as previously reported, but we also discovered very low levels of normally spliced transcript (**S2 Fig**). Comparison of experimental outcomes to predictions by *in silico* tools identified 12/15 exonic variants that were accurately predicted (**Table 1**), with one false negative incorrectly called by both algorithms and two variants deemed "indeterminate" due to a lack of consensus between the two tools (**S1 Table**).

### 'Missense' variants resulting in only misspliced mRNA isoforms may be misclassified as drug responsive

Given our observation that 6/15 exonic variants studied resulted in complete missplicing, we hypothesized that these 'missense' variants could be misclassified as modulator responsive if small molecule drugs were tested on the predicted protein isoforms containing an altered amino acid, as opposed to those generated by the misspliced transcripts. This was of particular concern for *CFTR* bearing the variant c.2908G>C (predicted effect G970R), which was shown to respond well to ivacaftor *in vitro*, whereas individuals with CF carrying this variant that were entered into a clinical trial did not respond [28, 45]. Indeed, c.2908G>C was recently reported to alter *CFTR* mRNA splicing in primary cells, thus explaining the lack of response [46]. Therefore, study of these variants also provided an opportunity to test the ability of EMGs to replicate what has been observed *in vivo*, and extend splicing analysis to an additional variant in this codon. To this end, we introduced three known CF-causing variants that affect this codon 970 into an EMG containing full-length intron 14, abridged intron 15, full-length intron 16, and abridged introns 17 and 18 (EMG_i14-i18, **Fig 2A**, left). To quantify the ratio of the *CFTR* mRNA isoforms, we performed RT-PCR using a FAM-6 tagged primer followed by fragment analysis. Interestingly, the c.2909G>A (predicted effect G970D) variant at the beginning of exon 18 had no effect on *CFTR* splicing (**Fig 2A**, right). In contrast, the two variants that alter the last nucleotide of exon 17, c.2908G>A (predicted effect G970S) and c.2908G>C (predicted effect G970R), resulted in missplicing and complete absence of full-length RNA transcript. For both of these variants, the change in the last nucleotide of the exon disrupted the 5' consensus splice donor site resulting in either complete skipping of exon 17 (**Fig 2A** left, isoform 1) or partial skipping of exon 17 (177-nucleotide in-frame deletion) through use of a cryptic splice donor within the exon (**Fig 2A** left, isoform 2). As compared to WT EMG_i14-i18 and c.2909G>A EMG_i14-i18, which both produced 100% normally spliced transcript, we found that c.2908G>C EMG_i14-i18 resulted in 78.5%±3.2 of *CFTR* transcript skipping exon 17 and 21.5%±3.2 of transcript harboring the 177-nucleotide deletion. In contrast, c.2908G>A EMG_i14-i18 favored the 177-nucleotide deletion isoform with 68.3%±2.7 of this product versus 31.1%±2.7 of the shortened isoform (**Fig 2A**, right).

Of the two splice isoforms observed, we would expect the exon 17 skipped isoform (isoform 1) to result in a frame-shift (Asn886Lysfs5Ter), while the partial exon skipping isoform (isoform 2) would result in an in-frame deletion (Ser912_Gly970del). We have previously shown that frameshifts in this region of the *CFTR* gene do not result in stable CFTR protein [38], however we wanted to verify that the 177-nucleotide deletion isoform would allow for production of shortened CFTR. To this end, we performed immunoblotting of protein lysates extracted from HEK293 cells transfected with EMG constructs. To assess CFTR protein processing in the absence of splicing effects, we also introduced each variant to a plasmid expressing *CFTR* cDNA. CFTR processing was assessed by comparing lower molecular weight, core

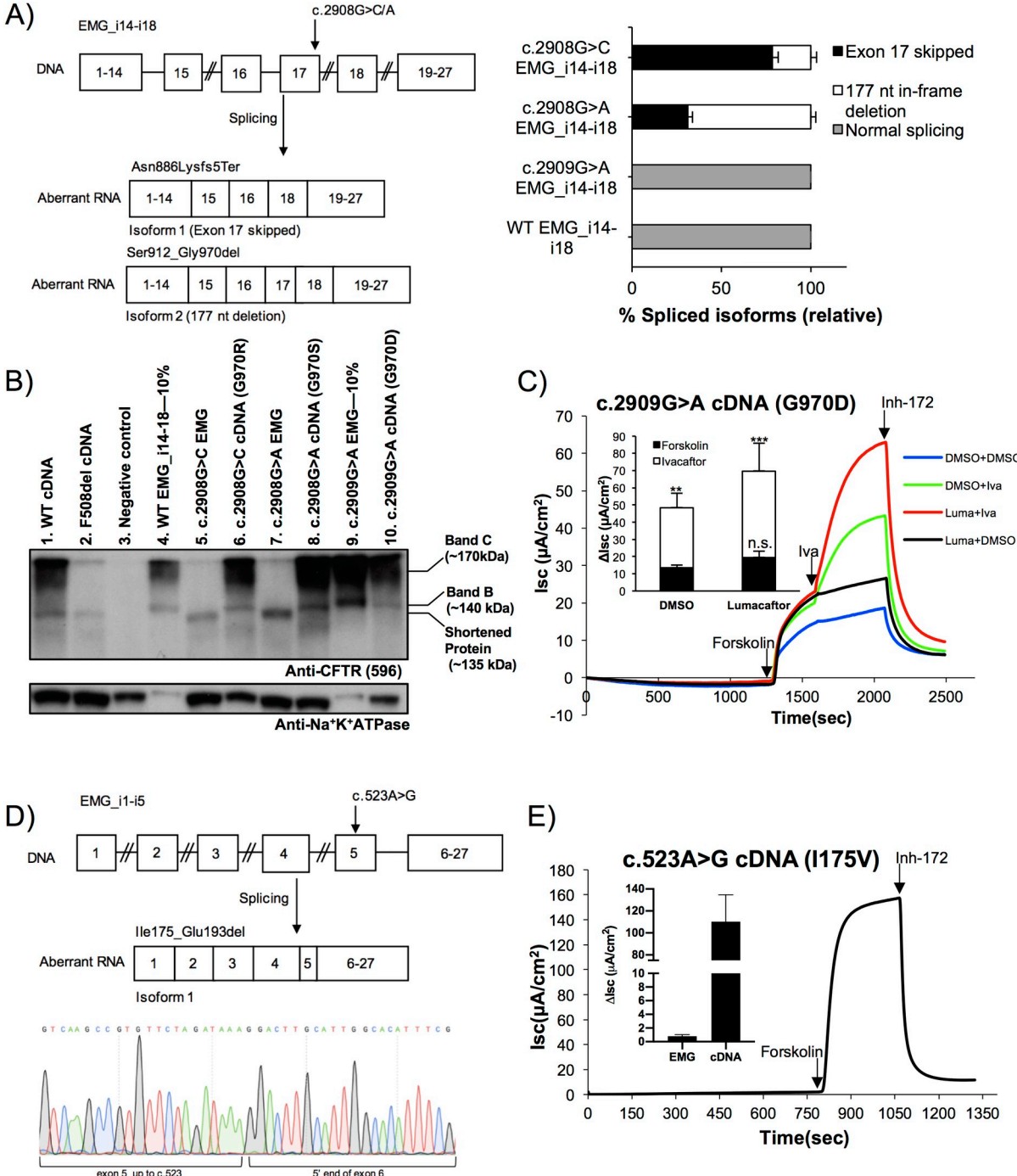

**Fig 2. Variants at codon 970 have different effects on mRNA splicing, protein processing and function of CFTR. (A) c.2908G>C (predicted effect G970R) and c.2908G>A (predicted effect G970S) alter mRNA splicing, while c.2909G>A (predicted effect G970D) allows synthesis of normally spliced *CFTR* mRNA. Left panel.** Schematic showing EMG_i14-i18 bearing c.2908G>A or c.2908G>C and *CFTR* mRNA isoforms produced. **Right panel.** Fragment analysis of RT-PCR products labelled with 6-FAM reveals the relative abundance of misspliced isoforms of *CFTR* transcript for EMG_i14-i18 bearing c.2908G>C or c.2908G>A. EMG_i14-i18 bearing c.2909G>A and wildtype (WT) EMG_i14-i18 served as controls. RNA isoform quantity was determined from the area under the curve (AUC) for each RT-PCR (minimum of three technical replicates per construct). The percent shown for each isoform is relative to the sum of the AUC for both isoforms. Error bars represent standard deviations. **(B) EMGs with c.2908G>C and c.2908G>A do not produce processed CFTR protein, while c.2909G>A does**. Immunoblot of protein lysates from transiently transfected HEK293 cells showing CFTR on top, with $Na^+,K^+$-ATPase on bottom as a loading control. Band C is mature, complex glycosylated CFTR protein. Band B is immature, core glycosylated CFTR protein. Negative control is an empty vector plasmid. WT cDNA, F508del cDNA, and WT EMG_i14-i18 served as controls. WT EMG_i14-i18

and EMG_i14-i18 bearing c.2909G>A were loaded at a reduced concentration (10%) as compared to other lysates to facilitate better visualization of less abundant isoforms in adjacent lanes **(C) G970D-CFTR generates reduced levels of chloride transport, which is augmented by ivacaftor and lumacaftor treatment.** Representative tracings for short circuit current ($I_{sc}$) assay of CFTR channel function and CFTR modulator response performed on CF bronchial epithelial (CFBE) cells stably expressing *CFTR* cDNA bearing G970D. Inset-Quantification of change in current ($\Delta I_{sc}$) in response to modulators (minimum of two independent measurements per condition). Data shown as mean±SD. *p* value determined by one-way ANOVA. *** ($p \leq 0.001$), ** ($p \leq 0.01$), n.s. (not significant, $p > 0.05$) when compared to DMSO treated vehicle control. **(D) The *CFTR* variant c.523A>G (predicted effect I175V) creates a cryptic donor splice site. Top panel.** Schematic showing EMG_i1-i5 bearing c.523A>G and resulting misspliced *CFTR* mRNA. **Bottom panel.** Sanger sequencing results confirm truncation of *CFTR* exon 5 at c.523 followed by the beginning of *CFTR* exon 6. **(E) *CFTR* cDNA bearing c.523A>G produces functional protein, while EMG_i1-i5 bearing the same variant produces no functional protein.** Representative tracing for short circuit current ($I_{sc}$) assay of CFTR channel function performed on CFBE cells stably expressing the c.523A>G cDNA construct. Inset- Comparison of quantification of change in current ($\Delta I_{sc}$) measured for CFBE cells stably expressing either c.523A>G EMG construct or c.523A>G cDNA construct. Data shown as mean±SD. Data underlying graphs in this figure reported in S3 Data.

glycosylated immature (band B) protein to the higher molecular weight, complex glycosylated mature (band C) protein. WT cDNA and WT EMG_i14-i18 controls produced predominantly mature CFTR, as well as some immature protein due to overexpression from a constitutive promoter (**Fig 2B**, lanes 1 and 4). As expected, F508del produced immature protein (**Fig 2B**, lane 2). Expression of cDNA (that does not require splicing) bearing c.2908G>C (G970R), c.2908G>A (G970S), or c.2909G>A (G970D), generated mature CFTR (**Fig 2B**, lanes 6,8,10). In contrast, c.2908G>C and c.2908G>A in an EMG (that requires splicing) produced protein of a lower molecular mass than immature, core glycosylated (band B) of their respective cDNA counterparts. This protein isoform is consistent with the predicted 59 amino acid deletion (p. Ser912_Gly970del) that partial exon skipping (177-nucleotide deletion) would produce and cannot be attributed to the exon 17 skipped isoform as the CFTR antibody recognizes an epitope downstream of where this isoform would be expected to truncate. As expected from our observation of normal splicing, c.2909G>A in an EMG produced mature CFTR (**Fig 2B**, lane 9).

To determine whether the mature CFTR protein bearing c.2909G>A (p.Gly970Asp) was functional, we created CF bronchial epithelial (CFBE) cell lines that stably expressed C.2909G>A cDNA from a single integration site as previously described [47]. Cells were grown in monolayers on filters to allow for polarization and mounted in Ussing chambers to assess CFTR chloride ion channel activity by measuring short circuit current ($I_{sc}$). Addition of forskolin was used to activate channel activity via cAMP-mediated signaling, followed by inhibition with the CFTR-specific inhibitor compound (Inh-172), allowing for quantification of the CFTR-specific change in current ($\Delta I_{sc}$±SD). We observed residual channel activity corresponding to approximately 3% of what we observe in our WT *CFTR* cDNA cell lines after normalizing for variation in expression (calculated as previously described, [30]). Treatment with the CFTR corrector, lumacaftor, for 24 hours resulted in a moderate increase in channel activity corresponding to a ~1.4-fold change ($\Delta I_{sc}$ = 19.55μA/cm$^2$ ±3.62 as compared to a DMSO treated control with $\Delta I_{sc}$ = 13.56μA/cm$^2$ ±1.54, **Fig 2C,** black line compared to blue line). Acute treatment with the CFTR potentiator, ivacaftor, also increased channel activity by about 3.6-fold ($\Delta I_{sc}$ = 48.5μA/cm$^2$ ±8.31, **Fig 2C,** green line). The combination therapy of 24hr lumacaftor treatment followed by acute ivacaftor treatment resulted in a greater increase than either modulator alone ($\Delta I_{sc}$ = 69.8μA/cm$^2$ ±16.1, **Fig 2C,** red line) with a ~5.1-fold increase over DMSO control. Together, these results indicate that only one of three variants at codon 970 predicted to substitute an amino acid allows normal RNA splicing and production of mutant CFTR protein that responds to modulators (c.2909G>A (G970D)). Previous studies have shown that individuals bearing c.2908G>C (G970R) do not respond to modulator treatment [45] and here we have shown that c.2908G>A (G970S) produces the same misspliced mRNA isoforms and would thus be expected to show no response.

All four of the 'middle' of exon variants were predicted and subsequently shown experimentally to also result in production of predominately misspliced mRNA isoforms. To assess the feasibility of modulator treatment for this type of variant, we performed an in-depth study of c.523A>G (predicted effect I175V). We were particularly interested in this variant because previous studies had indicated that individuals harboring c.523A>G have severe disease [48], while functional testing had an opposite result indicating high levels of function for CFTR protein with the I175V amino acid substitution [49]. We introduced c.523A>G to an EMG containing abridged introns 1–4 and full-length intron 5 (EMG_i1-i5, **Fig 2D**, top) to determine if missplicing could explain the discrepancy between these two reports, and if the outcome would impact drug response. Transcript sequencing revealed that c.523A>G activated a cryptic 5' splice site within exon 5 of *CFTR* that caused an in-frame deletion encompassing the last 57 nucleotides of exon 5 (**Fig 2D**, bottom). Normally spliced transcript could not be detected.

We then wanted to know if the shortened CFTR protein expected to be produced by c.523A>G (**S3 Fig**) would have any residual channel activity, and how this would compare to the function of CFTR protein harboring the I175V amino acid substitution if splicing was not considered. To this end, we created CFBE cell lines that stably expressed c.523A>G on the EMG background and on the cDNA background in order to perform functional testing. While the cDNA bearing c.523A>G generated robust CFTR chloride currents, cells expressing the c.523A>G EMG produced minimal current corresponding to <0.5% of CFTR function observed in a WT EMG_i1-i5 cell line (**S4 Fig**). Thus, missplicing explains the severe phenotype associated with c.523A>G. In contrast to these EMG results, CFBEs expressing c.523A>G on a cDNA background, representing the effect of the amino acid substitution alone (p.Ile175Val), produced robust forskolin activated and inh-172 inhibited current characteristic of CFTR (**Fig 2E**). The magnitude of current was comparable to about 38% of WT. Additionally, we found that CFTR bearing I175V was responsive to modulators (**S5 Fig**). These results show that the amino acid substitution I175V allows for production of modulator responsive CFTR, illustrating that study of this variant on a cDNA background would result in incorrect classification of this variant as modulator responsive.

## Residual normal splicing of an exonic variant produces functional, modulator responsive CFTR protein

To study the effect of exonic variants resulting in partial missplicing, we chose c.3873G>C (predicted effect Q1291H) located at the last nucleotide of *CFTR* exon 23. We were interested in this variant due to variability in disease severity between individuals harboring this variant and previous reports that it affects *CFTR* mRNA splicing *in vivo* [50]. Two *CFTR* mRNA isoforms are produced when c.3873G>C is introduced to an EMG containing abridged introns 21–24 (EMG_i21-i24, **Fig 3A**, left). RNA isoform 1 was a misspliced product utilizing a cryptic noncanonical GC splice donor site, resulting in retention of 29 nucleotides of *CFTR* intron 23 (**Fig 3A**, left). Of note, we also observed a second isoform that was consistent with normal splicing resulting in a full-length transcript harboring the missense change (**Fig 3A**, left). To determine the ratio of misspliced to normally spliced *CFTR* mRNA isoforms, we performed fragment analysis. As compared to WT EMG_i21-24, which produced 100% normally spliced transcript, we found that c.3873G>C on the EMG background resulted in 62.8%±2.2 misspliced transcript and 37.2%±2.2 normally spliced transcript (**Fig 3A**, right).

While our findings indicate that individuals with c.3873G>C would be expected to have reduced levels of *CFTR* transcript as a result of degradation of the misspliced product, we wanted to characterize the channel activity and drug response of the residual full-length protein that would harbor the Q1291H amino acid substitution. We found that cell lines

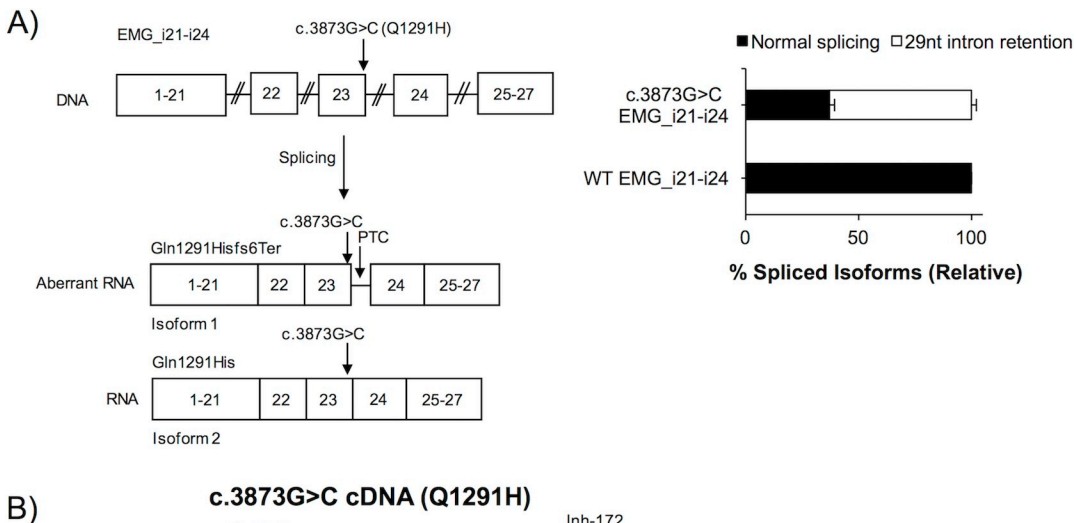

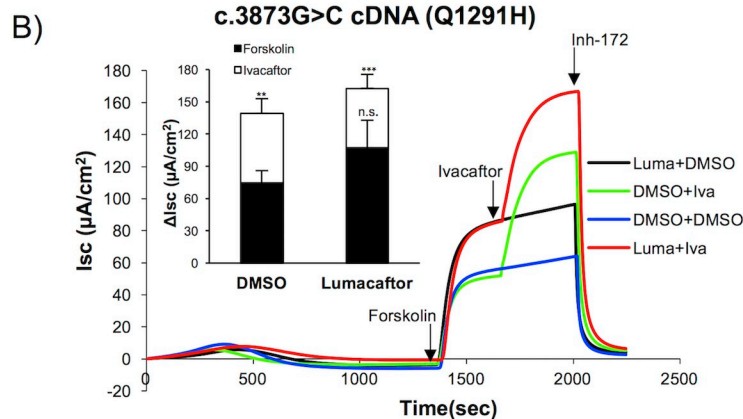

**Fig 3.** A variant at the end of exon 23 affects splicing of *CFTR* mRNA and causes an amino acid substitution in CFTR protein (A) The variant c.3873G>C (predicted effect Q1291H) weakens the canonical donor splice site resulting in partial intron retention and residual normally spliced *CFTR* transcript. Left panel. Schematic showing location of c.3873G>C relative to EMG_i21-i24 and resulting *CFTR* splice isoforms. Right panel. Fragment analysis of RT-PCR products labelled with 6-FAM reveals the ratio of misspliced to normally spliced *CFTR* transcript for EMG_i21-i24 bearing c.3873G>C. WT EMG_i21-i24 served as control. RNA isoform quantity was determined from the AUC (minimum of three technical replicates). The percent shown for each isoform is relative to the sum of the AUC for both isoforms. Error bars represent standard deviations. (B) Q1291H-CFTR is highly functional and responds to ivacaftor, lumacaftor, and combination therapy. Representative tracings for short circuit current ($I_{sc}$) assay of CFTR channel function and CFTR modulator response performed on CFBE cells stably expressing *CFTR* cDNA bearing c.3873G>C. Inset- Quantification of change in current ($\Delta I_{sc}$) in response to modulators (minimum of three independent measurements per condition). Data shown as mean±SD. *p* value determined by one-way ANOVA. *** ($p \leq 0.001$), ** ($p \leq 0.01$), n.s. (not significant, $p > 0.05$) when compared to DMSO treated vehicle control. Data underlying graphs in this figure reported in S4 Data.

expressing c.3873G>C cDNA showed high levels of CFTR-specific current. This was evidence that the amino acid substitution p.Gln1291His allows for proper folding, processing, and trafficking of CFTR protein, producing short circuit currents consistent with approximately 80% of what we observe in expression matched WT cDNA cell lines (calculated as previously described [30]). In addition, we found that treatment with any or multiple modulators further augmented channel function. Specifically, 24hr treatment with lumacaftor resulted in a ~1.4-fold change over baseline ($\Delta I_{sc}$ = 107.2μA/cm$^2$ ±25.7 compared to $\Delta I_{sc}$ = 74.5μA/cm$^2$ ±11 for DMSO control, **Fig 3B** black line compared to blue line), acute treatment with ivacaftor resulted in approximately a 1.9-fold change ($\Delta I_{sc}$ = 139.2μA/cm$^2$ ±13.8, **Fig 3B** green line), and combination therapy resulted in a ~2.2-fold change ($\Delta I_{sc}$ = 162.5μA/cm$^2$ ±12.9, **Fig 3B** red line). This indicates that the small amount of full-length transcript that is generated from

CFTR bearing c.3873G>C produces CFTR protein that is partially functional and targetable by modulator therapy.

## Variants in intronic splice sites can generate residual full-length wildtype transcript

Having shown that exonic variants can have unanticipated RNA-level effects, we next looked at the more classically considered splice affecting variants—those located within the introns. These variants, particularly those in the canonical and consensus splice sites, are often assumed to result in complete missplicing of the affected transcript, thus making them difficult-to-treat with protein-targeted therapies. To test this hypothesis, we chose 37 naturally occurring *CFTR* intronic variants to study in our EMG system. 19 were located in the canonical splice sites (+1, +2, -1, or -2) and 15 variants were located in the extended consensus splice sites (+3, +4, +5, and -3, "proximal intronic"). An additional three variants were located outside of the consensus splice sites ("distal intronic"), as previous studies have indicated that these can result in activation of cryptic splice sites and partial missplicing [37]. As with our exonic variants, all variants were assessed by both CryptSplice and SpliceAI. We utilized transient transfection and RT-PCR to evaluate the effect on *CFTR* mRNA splicing. We found that 19/19 canonical splice site variants resulted in production of one or more misspliced isoforms (**Table 2**). Interestingly, two variants (c.4005+2T>C, c.4242+2T>C) resulted in production of residual full-length transcript due to incomplete missplicng, which allowed for production of full-length CFTR protein (**S6 Fig**). Notably, both of these variants change the canonical GT donor site to a noncanonical GC, which is used to produce the full-length transcript. We found that 14/15 proximal intronic variants resulted in missplicing, while only one variant (c.164+3_164+-4insT) produced WT levels of full-length *CFTR* transcript. Importantly, five of these 14 misspliced variants allowed for production of residual normally spliced transcript. Of these 15 variants, 11 were accurately assessed by *in silico* tools with three "indeterminate" calls (due to lack of consensus between the two algorithms, **S1 Table**) and one false negative. Of the three distal intronic variants evaluated, only one (c.3717+40A>G) caused missplicing, consistent with predictions (**Table 2**). Overall, we identified eight intronic variants that could allow for production of residual WT CFTR (**Table 2**).

## Intronic variants that allow for residual full-length protein production are modulator responsive

Given our observation that 8/37 of our intronic variants allowed for production of residual full-length *CFTR* mRNA, we wanted to determine if these variants resulted in sufficient WT protein production to allow for residual channel function and modulator response. To test this hypothesis, we obtained primary nasal epithelial cells (HNE) from a CF individual with the genotype c.2657+5G>A/W1282X. To verify that our *in vitro* EMG results were recapitulated *in vivo*, RNA was obtained from these HNEs and sequencing was performed. We looked for the presence of reads mapping to erroneous exon junctions as visualized in a sashimi plot (**Fig 4A**). As compared to a WT control, we observed evidence of transcripts missing exon 16 with three reads mapping from exon 15 to exon 17 (**Fig 4A**). Notably, despite the presence of a nonsense variant on the other allele, 9 reads mapping from exon 15 to exon 16 were observed (compared to 27 reads mapping to this junction in a WT control), corroborating that this variant results in only partial missplicing. This finding was further validated by RT-PCR and sanger sequencing showing that the majority of the normally spliced transcript comes from the c.2657+5G>A allele (**S7 Fig**). To evaluate CFTR function in this individual, a well-differentiated monolayer culture of HNEs grown at air-liquid interface (ALI) was established on

**Table 2. Intronic variants evaluated for impact on *CFTR* mRNA splicing.**

| CANONICAL SPLICE SITE (+1, +2, -1, -2) | | | | |
|---|---|---|---|---|
| HGVS (Legacy) | Predicted Effect | Experimental Outcomes | | |
| Nucleotide change | Splicing (*in silico* prediction)* | RNA effect | Protein present | Modulator response |
| c.164+1 G>A (296+1 G>A) | Missplices | Missplices | Shortened | — |
| c.164+2 T>C (296+2 T>C) | Missplices | Missplices | Shortened | — |
| c.165-2 A>G (297–2 A>G) | Missplices | Missplices | Shortened | — |
| c.273+1 G>A (405+1 G>A) | Missplices | Missplices | Shortened | Non-responsive |
| c.274-1 G>A (406–1 G>A) | Missplices | Missplices | Shortened | — |
| c.274-2 A>G (406–2 A>G) | Missplices | Missplices | Shortened | — |
| c.489+1 G>T (621+1 G>T) | Missplices | Missplices | Shortened | — |
| c.579+1 G>T (711+1 G>T) | Missplices | Missplices | Shortened | Non-responsive |
| c.1584+1 G>A (1716+1 G>A) | Missplices | Missplices | Shortened | — |
| c.1585-1 G>A (1717–1 G>A)[36] | Missplices | Missplices | No protein | — |
| c.2658-1 G>C (2790–1 G>C) | Missplices | Missplices | No protein | — |
| c.2658-2 A>G (2790–2 A>G) | Missplices | Missplices | No protein | — |
| c.2988+1 G>A (3120+1 G>A)[36] | Missplices | Missplices | No protein | — |
| c.3469-2 A>G (3601–2 A>G) | Missplices | Missplices | Shortened | — |
| c.3717+1 G>A (3849+1 G>A) | Missplices | Missplices | Shortened | — |
| c.3718-1 G>A (3850–1 G>A) | Missplices | Missplices | Shortened | — |
| c.3873+1 G>A (4005+1 G>A) | Missplices | Missplices | Shortened | — |
| c.3873+2 T>C (4005+2 T>C) | Missplices | Missplices | Residual full-length | — |
| c.4242+2 T>C (4374+2 T>C) | Missplices | Missplices | Residual full-length | Responsive |
| PROXIMAL INTRONIC (+3, +4, +5, -3) | | | | |
| Nucleotide change | Splicing (*in silico* prediction)* | RNA effect | Protein present | Modulator response |
| c.164+3_164+4insT (296+3insT)† | Does not missplice | Does not missplice | Full-length | — |
| c.165-3 C>T (297–3 C>T) | Does not missplice | Missplices | Residual full-length | Responsive |
| c.273+3 A>C (405+3 A>C) | Missplices | Missplices | Shortened | — |
| c.489+3 A>G (621+3 A>G) | Missplices | Missplices | Residual full-length | — |
| c.579+3 A>G (711+3 A>G) | Indeterminate | Missplices | Shortened | — |
| c.579+3 A>C (711+3 A>C) | Missplices | Missplices | Shortened | — |
| c.579+3 A>T (711+3 A>T) | Indeterminate | Missplices | Shortened | — |
| c.579+5 G>A (711+5 G>A) | Indeterminate | Missplices | Shortened | — |
| c.2657+2_2657+3insA (2789+2insA)[36] | Missplices | Missplices | Residual full-length | Responsive |
| c.2657+5 G>A (2789+5 G>A)[36] | Missplices | Missplices | Residual full-length | Responsive‡ |
| c.3468+2_3468+3insT (3600+2insT) | Missplices | Missplices | No protein | — |
| c.3468+5G>A (3600+5 G>A) | Missplices | Missplices | Residual full-length | Responsive |
| c.3717+4 A>G (3849+4 A>G) | Missplices | Missplices | Shortened | — |
| c.3717+5 G>A (3849+5 G>A) | Missplices | Missplices | Shortened | — |
| c.3718-3 T>G (3850–3 T>G) | Missplices | Missplices | Shortened | — |
| DISTAL INTRONIC | | | | |
| Nucleotide change | Splicing (*in silico* prediction)* | RNA effect | Protein present | Modulator response |
| c.164+28 A>G (296+28 A>G)† | Does not missplice | Does not missplice | Full-length | — |
| c.2620-26 A>G (2752–26 A>G)† | Does not missplice | Does not missplice | Full-length | — |
| c.3717+40 A>G (3849+40 A>G)[37] | Missplices | Missplices | Residual full-length | Responsive |

The numbers shown in the superscript indicate previously published study on the respective variant.

Data supporting these findings in **S2 Data**.

*A prediction was deemed "indeterminate" if the two *in silico* tools used did not agree (S1 Table).

†This variant does not cause CF.

‡This variant is approved for modulator therapy.

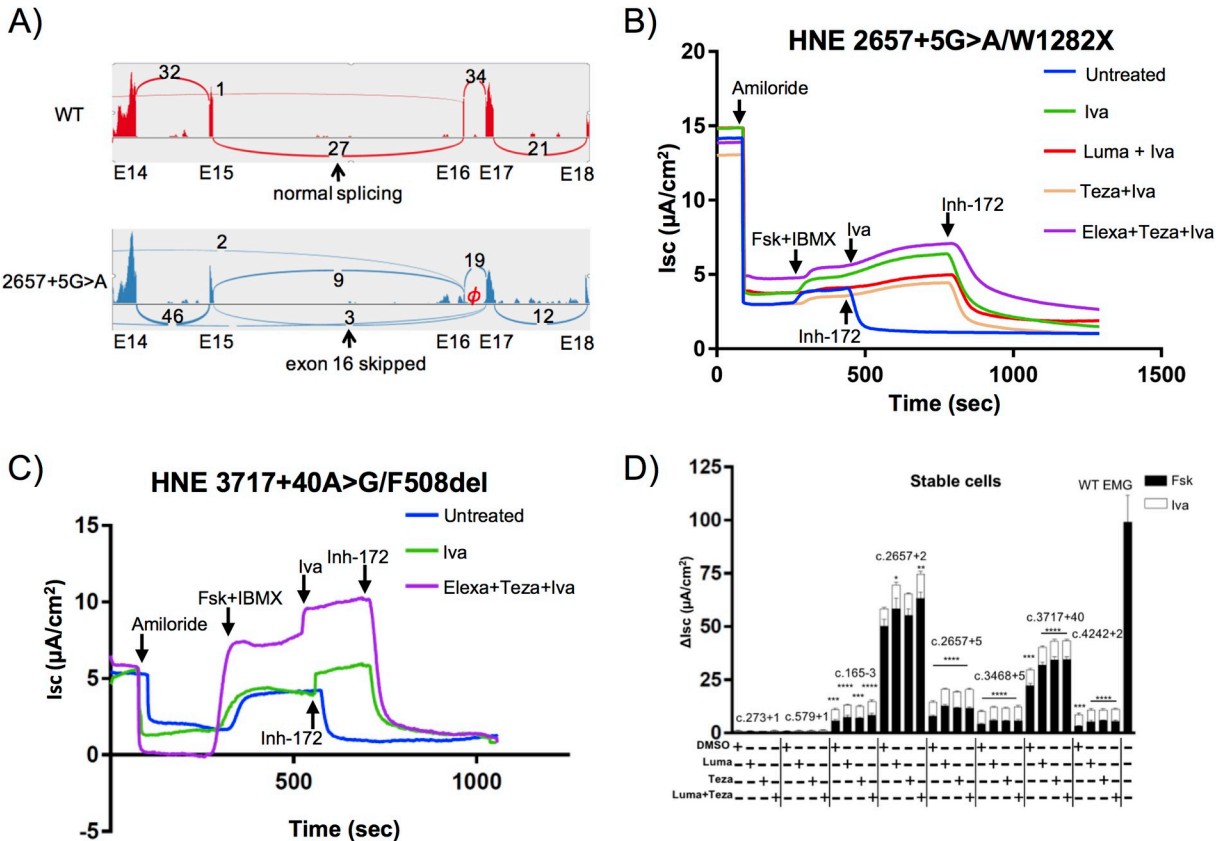

**Fig 4. CFTR intronic splice site variants with residual function are responsive to CFTR modulators. (A) Sashimi plots showing skipping of exon 16 caused by 2657+5G>A variant**. RNA-seq was performed on the total RNA extracted from nasal cells of the individual harboring c.2657+5G>A/W1282X genotype. RNA from healthy individual served as a control. Per-base expression is plotted on y-axis of Sashimi plot. 'E' refers to exon locations on x-axis. +5 on the Sashimi plot indicates variant is located in intron 16 (not to scale). Numbers on the Sashimi plot indicate counts of reads spanning exon junctions. $\phi$ on the illustration indicates relative location of the variant c.2657+5 G>A. **(B) Residual full-length *CFTR* transcript allows for modulator response in a c.2657+5G>A/W1282X individual**. Representative short circuit current ($I_{sc}$) tracings on Human nasal epithelial (HNE) cells harvested from an individual harboring c.2657+5G>A/W1282X. CFTR channel function and CFTR modulator response were evaluated in well-differentiated airway liquid interface (ALI) culture. **(C) Primary cells from an individual harboring c.3717+40A>G/F508del respond to modulator therapy.** Representative short circuit current ($I_{sc}$) tracings on Human nasal epithelial (HNE) cells harvested from an individual harboring c.3717+40A>G/F508del. CFTR channel function and CFTR modulator response were evaluated in well- differentiated airway liquid interface (ALI) culture. **(D) Intronic variants that allow for residual normal splicing respond to modulator therapies.** Stacked bar graphs indicate effect of modulator treatment on CFBE stable cells expressing different *CFTR* intronic variants. Change in current ($\Delta I_{sc}$) was defined as the current inhibited by Inh-172 after sustained $I_{sc}$ responses were achieved upon stimulation with forskolin alone or sequentially with ivacaftor. Data shown as mean±SEM (minimum of three independent measurements per condition). *p* value was determined by one-way ANOVA. ****($p \leq 0.0001$), n.s. (not significant, $p > 0.05$) when compared to DMSO treated vehicle control. Data underlying graphs in this figure reported in **S5 Data**.

snapwell filters and mounted in Ussing chambers to allow for $I_{sc}$ measurements to be taken. Amiloride was used to inactivate the apically located epithelial sodium channel (ENaC), followed by activation of CFTR via cAMP-mediated signaling with addition of forskolin and 3-isobutyl-1-methylxanthine (IBMX). As with CFBEs, we used inhibition with Inh-172 to allow for determination of the CFTR-specific change in current ($\Delta I_{sc}$ ±SEM). Assessment of channel activity confirmed that this residual normal splicing was sufficient to produce functional CFTR protein, as residual function ($\Delta I_{sc} = 2.9\mu A/cm^2$ ±0.12) was observed in untreated cells (**Fig 4B**, blue line). This function was further augmented by addition of modulator compounds. Notably, the greatest increase in function ($\Delta I_{sc} = 4.35\mu A/cm^2$ ±0.1), was observed when potentiator (ivacaftor) alone was added (**Fig 4B**, green line), corresponding to 30.6% of

what we observe, on average, in WT HNEs ($\Delta I_{sc}$ = 14.2µA/cm$^2$ ±4.0). Additionally, HNEs from a second individual who harbors the distal intronic variant c.3717+40A>G *in trans* with F508del also showed moderate levels of CFTR function ($\Delta I_{sc}$ = 3.86µA/cm$^2$ ±0.11 **Fig 4C**, blue line). This function could be attributed to partial missplicing caused by 3717+40A>G (**S8 Fig**) that leads to production of residual full-length protein [37]. As with c.2657+5G>A, ivacaftor increased 3717+40A>G-CFTR function by ~1.2-fold ($\Delta I_{sc}$ = 4.47µA/cm$^2$, **Fig 4C**, green line), consistent with our observation that ivacaftor elicits a response in WT HNEs (**S9 Fig**) and therefore could be acting on WT-CFTR produced by residual normally spliced transcript from the 3717+40A>G alleles. Interestingly, a combination of correctors (elexacaftor and tezacaftor) and potentiator (ivacaftor) resulted in the greatest increase in function ($\Delta I_{sc}$ = 8.72µA/cm$^2$, 61.3% of WT) an ~2.2-fold change, the majority of which could be attributed to the effect of these modulators on F508del-CFTR produced from the second allele (**Fig 4C**, pink line, [26]).

To investigate the modulator response of additional intronic variants, we stably expressed EMGs bearing eight intronic variants in CFBE cells (**Fig 4D**). Measure of CFTR channel chloride transport using $I_{sc}$ indicated that cell lines bearing variants shown to produce residual full-length transcript (c.165-3C>T, c.2657+2_2657+3insA, c.2657+5G>A, c.3468+5G>A, c.3717+40A>G, c.4242+T>C) have residual CFTR function, which can be augmented by treatment with corrector only, potentiator only, or combination therapy. Notably, cell lines bearing variants that cause complete missplicing and production of only shortened protein (c.273+1G>A, c.579+1G>T) did not demonstrate any residual channel activity, nor modulator response. These results show that intronic variants have to be experimentally tested to determine which allow for residual normal splicing, and thus production of WT CFTR protein that will respond to modulator therapies.

## Discussion

Splicing of mRNA is a critical process in the generation of protein, and disruption of this process has been implicated in a variety of genetic diseases [51]. Regulation of splicing is complex and involves both exonic and intronic sequences. In addition to the critical 5' and 3' consensus splice sites, regulatory sequences known as splice silencers and splice enhancers are found in both the exons (ESS, ESE, [52, 53]) and introns (ISS, ISE, [54, 55]). Additionally, interplay between RNA transcription and splicing as well as chromatin-level modulation add even more complexity [56]. Genetic variants can disrupt this process via many different mechanisms (e.g. weakening of canonical splice site, activation of cryptic splice site, disruption of splice regulatory region, etc)[57]. Through the study of 52 naturally occurring *CFTR* variants (exonic = 15, intronic = 37) we demonstrate the importance of delineating the specific molecular consequences of such variants for the accurate assignment of precision therapies. Our evaluation of exonic variants demonstrates the need to consider the RNA-level impact, as an amino acid substitution is not always the cause of disease. Additionally, we demonstrate that intronic variants, even within the consensus splice site, may retain some level of normal splicing opening up the possibility for protein-level treatment. These two overlooked mechanisms emphasize the importance of the role each specific nucleotide change plays in the manifestation of genetic disease.

By using heterologous expression systems to study individual *CFTR* variants, we were able to evaluate molecular mechanism of disease and drug response on an allele by allele basis and for rare variants for which primary cells are difficult to obtain. Previous work has established our expression minigene (EMG) system as a reliable model for the study of *CFTR* variants that impact mRNA splicing [34–38]. This approach utilizing minigenes harboring exons and the

adjacent native introns (either full-length or abridged) has also been undertaken to assess defects of mRNA splicing caused by disease-associated variants in different genes, *e.g.*, *PRPH2* [58], *CLRN1* (*USH3A*)[59], CNGβ1 [60]. A minimum of 200bp of intronic sequence from each end of the intron were used in generating abridged *CFTR* introns as this has been shown to be sufficient to capture the most critical splice regulatory sequences [61, 62].

One limitation of our model system is that *CFTR* transcription occurs outside of its native genomic context and only a portion of the gene sequence is present. While our constructs retain full exonic sequences and can thus account for exonic regulatory elements (e.g. composite exonic regulatory element of splicing in *CFTR* exon 12, [63]), our EMG studies cannot give insight into the effects of intronic sequences outside of the scope of our constructs on RNA splicing and processing. Transient expression of EMGs in HEK293s cannot account for the effects of chromatin, however stably integrated EMGs in our CFBE cell lines are subject to chromatin-level regulation, albeit at a different genomic locus than endogenous *CFTR*. While CRISPR/Cas and other methods of gene editing can generate cell lines expressing *CFTR* variants in the native genomic context [64, 65], EMGs provide a more rapid approach for assessing a large number of variants to identify those with unexpected consequences that warrant follow-up studies. Additionally, we show here and in prior work by ourselves and others that our results in EMGs are corroborated by studies in primary cells [20, 35–38, 44, 46, 50, 66].

Overall, we observed a pattern of missplicing consistent with previous observations regarding the degree of tolerance for variation at different exonic nucleotides within the consensus splice sites [6]. Namely, exonic variants that misspliced were more often located in the end of an exon as opposed to the beginning of an exon, consistent with the observation that the exon nucleotides of the 3' splice site are more tolerant of variation than the exon nucleotides of the 5' splice site [6]. Notably, we found that the splicing effects of 12/15 exonic variants were correctly predicted by the splice algorithms. Of the three not predicted, two were "indeterminate" (c.3873G>C, c.2909G>A) due to disagreement between the algorithms and were accurately assessed by SpliceAI, but not CryptSplice (**S1 Table**). One of these was a false negative for CryptSplice (c.3873G>C) and the other was a false positive (c.2909G>A). The false negative can be explained by the fact that the change in splice site strength sits just above the threshold for Cryptsplice. Additionally, c.3873G>A missplices through use of a noncanonical splice donor site that has a GC dinucleotide rather than a GT, which makes this outcome more difficult to predict. We have yet to identify an explanation for the false positive called only by CryptSplice, however our *in vitro* results recapitulate what others have reported in primary cells [46], giving additional evidence that the *in silico* prediction made by CryptSplice is incorrect. The variant which was incorrectly assessed by both algorithms (c.274G>A), resulted in only very low levels of missplicing which is likely why it was a false negative.

While the legacy names [41] of these exonic variants are given based on the presumed amino acid substitution, it is critical to note that only 3/9 misspliced variants allow for production of any of the predicted protein isoform. Thus, focusing on the amino acid substitution through *in vitro* studies in a cDNA based system can be misleading. In *CFTR*, a good example of this is the variant c.2908G>C (legacy G970R). This variant was evaluated *in vitro* and shown to respond robustly to modulator therapy [28]. On the basis of these data, individuals with CF who had at least one copy of this variant were enrolled in a clinical trial, but did not respond to therapy [45]. Studying this variant in the context of surrounding introns revealed the cause of this incongruence was an impact on *CFTR* mRNA splicing [46]. Here we corroborate these results in our EMG system providing further evidence that this is a considerable problem, but one which can be addressed by evaluating each nucleotide substitution and utilizing appropriate model systems. In addition, we show that the variant c.2908G>A (legacy G970S) also results in complete missplicing of *CFTR* mRNA. Importantly, our results replicate

prior studies showing that alteration of this amino acid at a different residue (c.2909G>A) does not have an impact on splicing, emphasizing the importance of considering variants at the nucleotide level. Prior studies in HNE cells from an individual with CF bearing c.2909G>A *in trans* with F508del showed evidence of modulator response [46] and our study of this variant in a heterologous system provides evidence that G970D-CFTR contributes to this response. Our study of the variant c.523A>G has revealed the potential for a similar misclassification of other splice-effecting variants outside of the consensus splice site. This variant appears to be of very high function when studied on a cDNA background, however the protein produced by the misspliced product (p.Ile175_Glu193del) does not retain any meaningful channel activity, which emphasizes the possibility of misclassification of disease liability and modulator response for exonic variants when relying on cDNA-based systems. Additional *CFTR* variants have been previously reported to follow this same paradigm of being named for an amino acid substitution that they ultimately do not produce [34, 37, 38, 44, 67]. Together these results suggest that exonic variants that missplice completely are unlikely to respond to modulators, while those that allow for production of some full-length protein have the potential to respond, albeit to a much lesser degree than would be expected if missplicing were not occurring. Additionally, so called 'synonymous' variants that would not be expected to result in a protein-level change are known to have RNA-level consequences. We have previously shown that the *CFTR* variant c.2988G>A, located in the last nucleotide of exon 18, results in partial missplicing [36] and a middle of exon 'synonymous' variant, c.2679G>T, has been reported to activate a cryptic splice site in *CFTR* exon 17 [68]. Importantly, these RNA-level effects of exonic variants have been reported in other disease-associated genes, such as *ATP7B* [69], *DMD* [15], and *PRPH2* [58], thus our work emphasizes the need for careful evaluation of molecular consequences of exonic variants across the genome.

When considering if a variant has been incorrectly assessed, a useful metric is a comparison of genotype and phenotype. This is well demonstrated by the example of c.523A>G. This variant was identified in individuals to cause severe disease [48], but shown *in vitro* to allow for production of highly functional protein [49]. The contrast between these two findings points to some overlooked mechanism, which we demonstrate here to be splicing. Similarly, high levels of variation in the phenotype of individuals harboring the same variant may be explained by a complex molecular mechanism of disease, such as missplicing. This is demonstrated by the example of c.3873G>C. Clinical features of individuals harboring this variant have a large amount of variability. For example, sweat chloride (a metric shown to correlate with disease severity, [22]) ranges from 35–112 mmol/L in these individuals (CFTR2). This may be explained by variability in the amount of normally spliced transcript between individuals, which would in turn impact the amount of residual CFTR function. Additionally, accurate assessment of disease liability and therapeutic options is particularly complicated for such variants that are exonic but also result in partial missplicing as the residual normal splicing results in a full-length protein that harbors an amino acid substitution. Assignment of precision therapies thus requires determination of both RNA and protein-level effects.

In addition to our evaluation of exonic variants, we chose to study 37 intronic variants spread across the *CFTR* gene and located throughout the introns. While +1, +2, -1, and -2 variants invariably caused missplicing, we observed more tolerance for variation in the +3, -3, and +5 positions consistent with previous reports [6]. Notably, these sites that were less likely to result in missplicing were also less accurately predicted by *in silico* tools, with our only false negative occurring at the -3 position (c.165-3C>T) as well as three "indeterminate" calls at the +3 and +5 positions (c.579+3A>G, c.579+3A>T, c.579+5G>A). Similar to our observation with c.274G>A (E92K), it is likely that the variant in the -3 position was not properly predicted because the major isoform is normally spliced. While SpliceAI outperformed

CryptSplice in evaluation of exonic variants, the three "indeterminate" predictions for intronic variants were the result of false negative calls by SpliceAI indicating that each algorithm has different strengths and weaknesses. Also of note, we identified two variants within the canonical splice site (c.3873+2T>C, c.4242+2T>C) that allowed for retention of some normal splicing. It has long been known that while less common than 5'GT donor sites, 5'GC donors are tolerated by the splicing machinery [7, 13]. Our findings indicated reduced splicing at these sites, consistent with a recent report which estimates that these splice sites can allow for production of up to ~80% of normally spliced transcript [8, 9].

Through evaluation of splice isoforms and CFTR channel function in primary human nasal epithelial cultures (HNE), we were able to demonstrate congruence between results determined *in vitro* (Table 2) and those determined *in vivo*. Our RNA-seq results show an alternative method of assessing RNA-level effects of variants. Additionally, the variant c.2657+5G>A is already approved for modulator therapy [24], thus showing that our EMG results and our HNE results can provide effective assessment of drug response. Previous studies have demonstrated that c.3717+40A>G allows for production of reduced levels of normally spliced transcript [37] and here we confirm that this could facilitate modulator response in individuals harboring this variant. While some of the response we observe in c.3717+40A>G can be attributed to the *in trans* F508del allele, complementary results in our heterologous expression system show that c.3717+40A>G likely contributes to this response. Additionally, c.2657+5G>A is *in trans* with a nonsense variant that underwent nonsense-mediated mRNA decay (NMD) (W1282X, **S7 Fig**) and thus we attribute the modulator response observed in HNEs to residual normally spliced transcript from the c.2657+5G>A allele. However, due to inter-individual variability in NMD efficiency some residual nonsense transcripts could be observed [70]. Although in our own studies, we have previously shown in primary cells that W1282X undergoes efficient NMD [39] and functional studies in W1282X homozygote HNEs have yielded no modulator response [71], which is further supported by a lack of clinical improvement in individuals treated with these drugs [72]. Therefore, our results for c.2657+5G>A confirm that modulator therapy can augment WT channel function when missplicing is incomplete and allows for production of residual WT CFTR. We also identified an additional six variants that responded, including c.3717+40A>G, which nicely corroborated our primary cell data. These results strengthen the validity of our *in vitro* studies and show that EMGs can be a useful tool in the characterization of variants and assignment of precision therapies.

Our systematic study of variants across the *CFTR* gene revealed that 43/52 variants (exonic = 9, intronic = 34) resulted in production of at least one misspliced mRNA isoform. We found that while *in silico* tools are useful for identifying splice variants, they are not failsafe and experimental validation is required to determine the degree of missplicing and the downstream functional consequences. Evaluation of these downstream impacts of missplicing was critical for assessment of residual function and modulator response. Both exonic and intronic variants that allow for production of protein should be assessed for treatment with modulator therapy. As modulator therapies for CF continue to improve, additional variants that allow for shortened protein isoforms to be generated may benefit from these types of treatments. However, some variants that result in complete missplicing may require alternative treatment strategies, such as direct modulation of splicing through modified snRNPs [73] or antisense oligonucleotides [74, 75] or correction of the primary genetic defect through gene editing approaches [76, 77]. Overall, we demonstrate the need for careful consideration of the molecular mechanism of a single nucleotide substitution when evaluating genetic variants for assignment of precision therapies. While we use CF here as a model for the study of splice variants, these variants are pervasive across the spectrum of genetic disease. Thus, our findings are a reminder that RNA-level impacts should be considered when assessing any genetic variant.

This becomes especially important when therapeutic approaches are dependent on targeting variants of a specific molecular mechanism.

## Materials and methods

### Ethics statement

This study was approved by the Institutional Review Board at Johns Hopkins Medicine, Baltimore (IRB# NA00029159 and IRB# 00116966). Written informed consent was obtained from all subjects.

### Assessment of variants by splice prediction tools

Middle of exon variants were chosen based on initial prediction by CryptSplice as previously described [37], and were then also assessed by the splice prediction algorithm SpliceAI using the recommended threshold of 0.5 [43]. All other variants were chosen to allow for evaluation of the impact on splicing in different regions of the gene and were assessed by CryptSplice and SpliceAI after selection. A prediction for a variant (**Tables 1 and 2**) was considered "indeterminate" if the two algorithms did not agree. Predictions for each variant with each algorithm can be found in supporting information (**S1 Table**).

### Introduction of variants to EMG and cDNA constructs

Expression minigene (EMG) plasmids were created as previously described [36, 38]. Variants were introduced to either EMG or cDNA plasmid using site-directed mutagenesis (SDM) as previously described [30, 36]. Briefly, plasmids were PCR-amplified using primers containing the desired single nucleotide change (**S2 Table**) and successful addition of the variant was verified by sanger sequencing.

### Transfection of HEK293 cells and evaluation of mRNA splicing

Plasmids were transiently introduced to HEK293 cells using Lipofectamine 2000 (Thermo-Fisher Scientific). Total RNA was collected 48hours post-transfection and 500ng was used as an input for reverse transcription, which was performed using iScript cDNA synthesis kit (BioRad). RT-PCR was then performed as previously described [36]. Briefly, 2μl of cDNA was used as input for PCR amplification using exonic primers spanning the region surrounding relevant introns and KOD Hot Start polymerase master mix (Millipore Sigma). Conditions for PCR were 2 minutes at 95˚C followed by 35 cycles of 20 seconds at 95˚C, 10 seconds at the annealing temperature (specific to each set of primers), and 20 seconds at 70˚C. PCR products were analyzed by gel electrophoresis followed by extraction of relevant bands and Sanger sequencing.

### Fragment analysis

Fragment analysis was performed as previously described [36]. Briefly, RT-PCR was performed as described above, with a forward primer bearing a 5'6FAM tag. Each RT-PCR was run in triplicate and relevant bands were gel extracted and purified. Products were separated on Applied Biosystems 3730 DNA Analyzer capillary electrophoresis system with GeneScan-500 Rox (Applied Biosystems) used as an internal size standard. Relative RNA isoform quantity was determined by the area under the curve (AUC) for each isoform compared to the total AUC for both isoforms for each sample and these values were then averaged across the technical replicates.

## Immunoblotting

Protein lysates were collected 48 hours after transient transfection of EMG or cDNA plasmids into HEK293 cells. 40μg of lysate were loaded per sample into a 7.5% Criterion TGX protein gel (BioRad). Transfer to PVDF membrane was performed in a Trans-Blot Turbo Transfer System (BioRad). After blocking, the membrane was probed with either mouse monoclonal anti-CFTR antibody 596 binding amino acids 1204–1211 (CFFT, University of North Carolina Chapel Hill) or mouse monoclonal anti-CFTR antibody 570 binding amino acids 731–742 (CFFT, University of North Carolina Chapel Hill) diluted to 1:5,000. Rabbit monoclonal anti-sodium/potassium-ATPase (Abcam) diluted 1:50,000 was used as a loading control. Secondary antibodies were anti-mouse (1:150,000 GE Healthcare) and anti-rabbit (1:100,000 GE Healthcare), respectively. Blots were exposed on film using ECL Primer Western Blotting Detection Reagent (GE Healthcare).

## Creation of stable cell lines

CF bronchial epithelial cells stably expressing EMG or cDNA constructs were generated as previously described [30, 38, 78]. Briefly, CFBEs lacking endogenous *CFTR* expression (CFBE41o-) and containing a single Flp recombinase target site [47] were co-transfected with EMG or cDNA plasmid and pOG44 (a plasmid encoding Flp recombinase) using lipofectamine LTX (ThermoFisher Scientific). Cells were placed under Hygromycin selection until individual clones grew large enough to isolate and propagate into new cell lines. For cDNA cell lines, genomic DNA was extracted from each clone and PCR amplification with overlapping primer sets spanning the entirety of *CFTR* was used to verify integration. For EMG cell lines, RNA was extracted and reverse transcribed into cDNA, which was then used for integration PCR. For all cell lines, *CFTR* expression level was verified by qPCR as previously described [30].

## Collection and culture of primary HNEs

Primary nasal epithelial cells (HNEs) were collected and cultured as previously described [38, 79–81] under Johns Hopkins University IRB#NA00029159 and IRB#00116966. An anesthetic was applied to the nasal mucosa proceeding brushing of the mid-part of the inferior turbinate. Nasal cells were harvested from both nostrils. Brushes were washed with propagation media as cells were scraped off the brush and the cells and media were collected and placed in a conical tube and pelleted by centrifugation. Brushes were then placed in PBS and vortexed to collect additional cells and this PBS was used to wash the pelleted cells. The cell pellet was then resuspended in 3mL Accutase and placed at 37°C for three minutes. Accutase was inactivated by the addition of propagation media (S3 Table) and cells were pelleted and resuspended in HNE propagation media (S3 Table). A flask containing a fibroblast feeder layer (3T3-J2, Kerafast) was irradiated in advance in the presence of 30 Gy by setting the CIXD Biological Irradiator at 220V, 13A, and 468 sec. Resuspended cells were added to the feeder layer flask and maintained in propagation media in the presence of 10μM Reagent Y (ROCK inhibitor) (S10A Fig). To facilitate differentiation, 300,000–400,000 cells were moved to snapwell filters coated with collagen type IV (Sigma#C6745-1ML) 0.3 mg/mL, final concentration 50 μg/mL. Cells were kept in propagation media for 5–7 days and upon confluency changed to either differentiation media containing Ultroser G or proprietary PneumaCult ALI media, both of which lack ROCK inhibitor (S3 Table). One day later, apical media was removed to establish air-liquid interface (ALI) culture. To maintain ALI culture, basolateral media was changed twice per week and the apical side of the filter was washed with 37°C PBS once per week. Prior to

functional assessment, cells were grown on ALI culture for 28 days, which is the time required for the appearance of cilia as a marker of complete differentiation (**S10B Fig**).

## Functional assessment and modulator testing

Assessment of CFTR channel function and response to drugs was performed in CFBEs as previously described [30, 47]. Briefly, CFBE stable cell lines were plated on snapwell filters and grown until transepithelial resistance reached ~200μΩ. Filters were mounted in Ussing chambers (Physiological Instruments). A high chloride solution was added to the basolateral chamber and a low chloride solution was added to the apical chamber. After equilibration of currents, 10μM forskolin (Selleckchem) was added to the basolateral side to activate CFTR channels via cAMP signaling. Currents were allowed to plateau and CFTR was inhibited using 10μM Inh-172 (Selleckchem) added to the apical chamber. Modulator testing for cDNA cell lines was carried out with addition of 6μM lumacaftor (Selleckchem) or equivalent volume DMSO 24hrs prior to run and acute apical addition of 10μM ivacaftor (Selleckchem) or equivalent volume of DMSO following plateau after channel activation. For EMG cell lines, correctors (lumacaftor or tezacaftor (Selleckchem)) were added individually or in combination 24hrs prior to run to a final concentration of 3μM. 10μM ivacaftor was added acutely. For all Ussing chamber studies the drop in current produced by the addition of Inh-172 (ΔInh-172) was used to quantify CFTR channel function.

Assessment of CFTR channel function and response to drugs was performed in primary HNEs as previously described [38]. Briefly, well-differentiated HNEs (**S10B Fig**) on ALI culture were mounted in Ussing chambers and bathed in a chloride solution. After equilibration of currents, 100μM of amiloride (Selleckchem) was added to the apical chamber to inactivate the epithelial sodium channel (ENaC). 10μM forskolin and 100μM 3-isobu-tyl-1-methylxan-thine (IBMX, Sigma-Aldrich) were added simultaneously to the basolateral chamber to activate chloride transport through cAMP-mediated channel opening. Specific inhibition of CFTR driven chloride transport was achieved through apical addition of 10μM Inh-172. WT HNE function was determined using an average of values obtained from two unrelated individuals with a total of 11 readings. Modulator testing was performed by 24hr addition of 3μM corrector (lumacaftor, tezacaftor, or tezacaftor+elexacaftor) and acute apical addition of 10μM ivacaftor.

## RNA-sequencing

RNA-sequencing preparation and analysis was performed as previously described [38]. Briefly, RNA was extracted from cultured primary HNEs and 1.0μg was used as input for library preparation. 50 million paired end reads were obtained and aligned to the reference genome (hg19) using BowTie2 [82]. Tophat2 [83] was used to determine splice junctions. To allow for visualization of splicing, sashimi plots were generated using the Integrative Genomics Viewer.

## Statistical analysis

Statistical analysis was completed using GraphPad Prism (GraphPad Software, San Diego, California USA, www.graphpad.com). One-way ANOVA was performed followed by Dunnett's test for multiple comparisons. A $p$-value of $<0.05$ was considered significant. Raw data underlying all graphs reported in **S3**–**S6 Data**.

Web resources

CFTR2 (Clinical and Functional Translation of CFTR), https://cftr2.org (version January 10, 2020)

CFTR mutation database, www.genet.sickkids.on.ca

CryptSplice, https://bitbucket.org/jhucidr/cryptsplice
Splice AI, https://github.com/Illumina/SpliceAI

## Supporting information

**S1 Fig. The beginning of exon variant c.274G>A (predicted effect E92K) results in a low level of exon 3 skipping. Top panel**. Representative raw data for fragment analysis of RT-PCR products from evaluation of splicing in HEK293 cells after transient transfection with EMG_i1-i5 bearing c.274G>A. Results show majority normally spliced transcript and a small fraction of products consistent with the size of exon 3 skipped transcript. **Bottom panel.** Representative raw data for fragment analysis of RT-PCR products from WT EMG_i1-i5 showing normally spliced transcript.
(TIF)

**S2 Fig. In an individual bearing two copies of c.3700A>G (predicted effect I1234V). c.3700A>G activation of a cryptic splice site results in production of primarily misspliced transcript. Left panel.** Fragment analysis of RT-PCR products shows abundance of normally spliced transcript relative to misspliced transcript. **Right panel**. Screenshot of IGV view showing RNA sequencing reads mapping to region surrounding c.3700A>G. Reduced coverage is evidence of misspliced product, while one read maps to exon/exon junction indicating some normal splicing.
(TIF)

**S3 Fig.** *CFTR* **cDNA bearing c.523A>G (175V) produces processed protein, while EMG_i1-i5 bearing the same variant produces only unprocessed CFTR.** Immunoblot showing CFTR on top, with $Na^+,K^+$-ATPase on bottom as a loading control. Lanes from a single blot were reordered and split into two panels to allow for appropriate comparison of matched experimentals and controls. Left panel. All constructs driven by EF1α promoter and express *CFTR* cDNA. WT and F508del served as controls. Negative control is an empty vector plasmid. Right panel. All constructs driven by CMV promoter and express either *CFTR* cDNA or *CFTR* EMG_i1-i5, as indicated. WT cDNA, F508del, WT EMG_i1-i5 served as controls.
(TIF)

**S4 Fig. CFBEs stably expressing WT EMG_i1-i15 produce substantial current.** Representative tracing for short circuit current ($I_{sc}$) assay of CFTR channel function performed on CF bronchial epithelial cells stably expressing the WT EMG_i1-i5 construct.
(TIF)

**S5 Fig. I175V-CFTR responds to modulator therapy.** Representative tracings for short circuit current ($I_{sc}$) assay of CFTR channel function and CFTR modulator response performed on CFBE cells stably expressing the I175V cDNA construct. Inset- Quantification of change in $I_{sc}$ in response to modulators (minimum of three independent measurements per condition). Data shown as mean±SD. *p* value determined by one-way ANOVA. *** ($p \leq 0.001$), *($p \leq 0.05$) when compared to DMSO treated vehicle control. Data underlying graph in this figure reported in **S6 Data**.
(TIF)

**S6 Fig. Residual normal splicing and protein production resulting from c.4242+2T>C variant in the CFTR terminal intron. (A)** Schematic illustration of the Expression minigene i25-i26 and location of the splice-site variant. Full-length introns 25 and 26 were inserted into pcDNA5FRT-CFTR cDNA construct to create the expression minigene. Variant c.4242 +2T>C was created by site directed mutagenesis. Two isoforms (aberrant and normal indicated on the labels) were observed in HEK293Flp cells by transient transfection of the EMG

harboring the c.4242+2T>C variant. **(B)** Sanger sequencing confirmed skipping of exon 26 in the aberrant spliced isoform resulting in frameshift and introduction of premature termination codon. The second minor isoform confirmed normal splicing. **(C)** Immunoblotting of the protein lysates collected from HEK293Flp cells transfected with either WT EMG_i25-26, F508del cDNA or c.4242+2T>C. Non-transfected parental cells served as control. CFTR antibody 596 (CFFT) was used to probe for CFTR protein production. Residual levels of full-length mature and immature protein products denoted as band C and B respectively were produced by c.4242+2T>C variant due to normal splicing. A lower sized truncated CFTR protein fragment resulting from aberrantly spliced isoform was also observed.
(TIF)

**S7 Fig. Residual level of normal spliced wild-type mRNA produced from c.2657+5G>A allele is higher than W1282X mRNA in the primary nasal epithelial cells of individual harboring c.2657+5G>A and W1282X in compound heterozygosity. (A)** Top panel, schematic illustration of the region selected to amplify *CFTR*. Vertical arrow indicates location of W1282X variant in the context of processed mRNA. Horizontal arrows indicate forward and reverse primers selected from exon 22 and exon 24 respectively for the reverse transcription-polymerase chain reaction (RT-PCR). Bottom panel, shows selection of primers to amplify TATA box binding protein (*TBP*) gene as control. (B) Ethidium bromide—stained agarose gel to visualize RT-PCR products. RNA was extracted either directly from the nasal cells dislodged from brush using a forcep (non-cultured) or nasal cells expanded in propagation medium containing 10 μM reagent-Y (see methods for culture and S3 Table for recipes). 200 ng RNA was used to prepare cDNA. PCR was performed on either stock cDNA or diluted (1:5). cDNA prepared from the non-CF cultured nasal cells was used as a positive control. The molecular weight of the amplification product matched the expected size products for *CFTR* (417 base pairs) and TBP (108 base pairs). Faint amplification was achieved for stock cDNA prepared from the brush, and very faint amplification for 1:5 diluted cDNA. TATA box binding protein (TBP) was amplified as control for the quality of RNA. No-RT, used as negative control, contained RNA from the cultured nasal cells of individual with 2789+5/W1282X. (C) Representative electropherogram of the RT-PCR product to assess differential expression of mRNA produced from 2789+5 and W1282X alleles. RT-PCR products obtained were sent for Sanger sequencing. Small peak for "nucleotide A" (indicated by vertical orange arrow) corresponding to W1282X allele, and large for "nucleotide G" corresponding to normal spliced wild-type mRNA produced from 2789+5 were observed. RT-PCR result shown here lends support to RNA-seq data that 2789+5 results in partial missplicing, and corroborates with residual CFTR function in the nasal cells observed by short-circuit current measurements.
(TIF)

**S8 Fig. RNA-sequencing performed on HNEs bearing c.3717+40A>G/F508del shows evidence of missplicing.** Top panel. Sashimi plot shows retention of 40 nucleotides of intron 23 as well as normal splicing. WT HNEs served as control. Numbers indicate number of reads mapping to each splice isoform. Bottom panel. Zoomed in version of sashimi plot for better visualization of intron retention.
(TIF)

**S9 Fig. Assessment of CFTR function and ivacaftor response in WT human nasal epithelial (HNE) cells.** Upper graph, two replicates of short-circuit current ($I_{sc}$)tracings from the same sample. Lower graph, same $I_{sc}$ tracings as above, but after amiloride inhibition was set to zero to allow for better visualization of the function.
(TIF)

**S10 Fig. Primary human nasal epithelial (HNE) cells at propagation and differentiation stages. (A)** Compound microscopy image of HNE cells propagating in conditionally reprogrammed co-culture with irradiated J2 feeder cell (J2s) at passage 0. Arrow indicates HNE cells growing in island surrounded by J2 fibroblast. **(B)** Scanning electron microscopy image of well- differentiated 28-days old HNE cells growing in air-liquid interface (ALI) culture. Cells propagating at passage 2 were transferred on to the filters to establish ALI. Arrow indicates appearance of cilia as a marker of differentiation.
(TIF)

**S1 Table. Splice predictions using bioinformatics tools.**
(XLSX)

**S2 Table. Site-directed mutagenesis (SDM) primers.**
(XLSX)

**S3 Table. Reagents used in culturing primary HNEs.**
(XLSX)

**S1 Data. Additional Sanger sequencing of RT-PCR products S1 Table.**
(PDF)

**S2 Data. Additional immunoblots S2 Table.**
(PDF)

**S3 Data. Raw data used in generating graphs in Fig 2.**
(XLSX)

**S4 Data. Raw data used in generating graphs in Fig 3.**
(XLSX)

**S5 Data. Raw data used in generating graphs in Fig 4.**
(XLSX)

**S6 Data. Raw data used in generating graph in S5 Fig.**
(XLSX)

## Acknowledgments

EMG containing introns 14 and 16 was a kind gift from Prof. Margarida Amaral and Dr. Anabela Ramalho, Faculty of Sciences, BioFIG—Centre for Biodiversity, Functional and Integrative Genomics, University of Lisboa, Lisboa, Portugal.

## Author Contributions

**Conceptualization:** Anya T. Joynt, Garry R. Cutting, Neeraj Sharma.

**Data curation:** Karen S. Raraigh.

**Formal analysis:** Anya T. Joynt, Neeraj Sharma.

**Funding acquisition:** Calvin U. Cotton, Garry R. Cutting, Neeraj Sharma.

**Investigation:** Anya T. Joynt, Taylor A. Evans, Matthew J. Pellicore, Emily F. Davis-Marcisak, Calvin U. Cotton, Natalie E. West, Christian A. Merlo, Garry R. Cutting, Neeraj Sharma.

**Methodology:** Anya T. Joynt, Taylor A. Evans, Matthew J. Pellicore, Emily F. Davis-Marcisak, Melis A. Aksit, Alice C. Eastman, Kathleen C. Paul, Derek L. Osorio, Alyssa D. Bowling, Calvin U. Cotton, Neeraj Sharma.

**Resources:** Shivani U. Patel, Natalie E. West, Christian A. Merlo, Garry R. Cutting, Neeraj Sharma.

**Software:** Melis A. Aksit, Alice C. Eastman.

**Supervision:** Garry R. Cutting, Neeraj Sharma.

**Validation:** Anya T. Joynt, Taylor A. Evans, Matthew J. Pellicore, Emily F. Davis-Marcisak, Neeraj Sharma.

**Writing – original draft:** Anya T. Joynt.

**Writing – review & editing:** Anya T. Joynt, Alyssa D. Bowling, Karen S. Raraigh, Garry R. Cutting, Neeraj Sharma.

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
