## [Decision Letter · Decision Letter 0]

4 Aug 2020

Dear Dr Sharma,

Thank you very much for submitting your Research Article entitled 'Evaluation of both exonic and intronic variants for effects on RNA splicing allows for accurate assessment of the effectiveness of precision therapies' to PLOS Genetics. Your manuscript was fully evaluated at the editorial level and by independent peer reviewers. The reviewers appreciated the attention to an important topic but identified some aspects of the manuscript that should be improved.

You will note that one reviewer felt additional germline editing experiments are necessary. While we disagree that such experiments are required, we do believe that some commentary addressing this point, including strengths, weaknesses, and limitations, would be valuable.  If there are any data available, such as examples of variants tested multiple ways that provide similar or discordant results, that would be ideal, but would not be required.

We therefore ask you to modify the manuscript according to the review recommendations before we can consider your manuscript for acceptance. Your revisions should address the specific points made by each reviewer.

[LINK]

Yours sincerely,

Gregory M. Cooper, PhD

Associate Editor

PLOS Genetics

Hua Tang

Section Editor: Natural Variation

PLOS Genetics

Reviewer's Responses to Questions

**Comments to the Authors:**

Reviewer #1: The paper from A.T .Juynt et al ,is devoted to the evaluation of both exonic and intronic variants for their effect on RNA splicing and for acurate assessement of the effectiveness in the context of precision therapies .

The CFTR gene, responsible of cystic fibrosis, when both alleles are mutated is a perfect model to decipher the relationship between a genetic variant and its fonctional effect both in vitro and in vivo .

This is particularly important in the field of cystic fibrosis as more than 2000 mutations or variants have been reported in this gene and since ten years new drugs called modulators (potentiators and correctors) have been discovered and are now proposed to CF patients in the context of allele specific efficiency.

By analysis individual CFTR variants they carefully determine the molecular mechanism associated with the change at the nucleotide level and the drug response on an allele by allele basis.

In the first part of their results they first selected missenses CFTR variants for evaluating their effect on mRNA splicing .To do that they chose naturally occuring CFTR variants located in the first and the second nucleotide of an exon as well as the last and the penultimate nucleotide .Moreover they selected also the splice –defect potential of CFTR missense variants predicted by the CryptSplice algorithm.They selected 15 variants for this study ,14 of these were introduced into expression minigenes (EMGs) that contained full length or abridged CFTR intronic sequences allowing the evaluation of the effects of CFTR variants on m RNA splicing of adjacent introns.

This first part is followed by showing carefully that missense variants could be misclassified as drug responsive ,then they showed that residual normal splicing of amissense variant produces functional ,mudulator responsive CFTR protein and finally they showed that variants in intronic splice sites can generate reduced level of full length wildtype transcript .

These data are extremely well performed on 52 naturely occuring variants combining in vitro ,in vivo and functional results .Doing that they demonstrate the need to consider the impact on RNA-level showing that an amino acid sbstitution is not always the cause of the disease and that intronic variants even in a consensus splice signal may retain some level of normal splicing allowing the possibility for protein –level treatment .

I think this is an excellent work that could be helpful not only for geneticists working in the field of CF but also for those working in the field of other genetic disorders

Reviewer #2: PLOS Genetics CF

Joynt and colleagues functionally characterize 52 variants of CFTR in a splicing assay. The majority are intronic, but they find exotic and intronic variants that alter splicing and confirm this in CFTR current assays and in primary cells from patients. They interpret their data in light of CF modulator therapies and provide mechanistic support for variants that respond or don’t respond to modulators. One interesting example was show in figure two where the different mutations in the same codon can causes different ratios of RNA species to be produced.

This paper’s study of a large number of variants leads, in my reading of this, to two key findings that should be of broad interest. First is that exonic variants can have “unanticipated RNA-level impacts”, and that genotype specific therapies for CF patients need to consider broader molecular mechanisms. The first finding I would not consider interesting, but not unanticipated, as previous reports have found exonic variants that are pathogenic due to altered protein binding to DNA vs. altered amino acid sequences, and likewise missense exonic variants near splice sites are routinely considered in light of splicing. The second finding currently largely applies to CF where identifying and expanding genotype specific therapies have made outstanding progress, but one hopes can be used as a model for other diseases, where The authors could broaden the interest by emphasizing the carful molecular assessment (i.e. splicing) of all variants is necessary as more diseases move into genotype driven precision therapies.

Minor Concerns

1. At various points it is very hard to figure out what they did and to how many variants. They authors should use consistent language, for example they switch from exonic, to missense to “variants of interest” within the same paragraphs on page 7 and 8. As I believe all variants tested are of interest, and exonic ones tested are missense, while all intronic ones are not, I would prefer they simplify and use “exonic” or “intronic” unless necessary to denote a functional consequence they uncover.

2. More description or details on CF for those not PLOS Genetics readers well versed in the disease. For example on #285 “We were interested in this variant due to its varying clinical consequences” they can describe the varying clinical consequences, which are described only as a broad sweat chloride range on line #511.

3. Fig 3B, Add data for WT CFTR so we can see how close is Luma+Iva to WT CFTR. The text states “approximately 80% of WT function” for Gln1291His, but this is not cited in any way and it’s unclear if this is a protein level/single channel measurement, or a measurement from cells genetically homozygous for 3873G>C.

4. Figure 4 showing data in HBE cells from a patient is a nice validation of what they see in EMG assays. I know procuring patient samples for rare genotypes is difficult, but any additional assays in primary cells would greatly strengthen this papers reliance on the EMG assays.

Reviewer #3: Comments to Author:

The manuscript by Joynt et al describes a need to experimentally validate exonic and intronic substitutions within the CFTR gene for effects on splicing, as this may be overlooked, especially at nonsynonymous substitutions near splice sites. This is especially important for proper characterization of CFTR variants in terms of prescription of relevant CF drugs. This type of analysis may also explain why genotype (and therefore predicted functional change) might not align with expected outcomes in drug studies for particular variants. The authors have used an array of previously designed tools/models to explore the effects of variants on splicing, primarily using expression minigenes (EMGs) containing relevant CFTR intronic sequences, and cDNA constructs to generate stable cell lines or primary HNEs with relevant genotypes in electrophysiology studies to measure channel activity and response to CFTR modulators. The results presented here are an extensive survey of the potential for mischaracterization of CFTR variant impact on CFTR function, however there important concerns that arise from this study as outlined below.

Major comments:

Though the data presented are no doubt the result of a substantial experimental effort, this paper is more suitable for Molecular Diagnostics Journal than PLOS Genetics, primarily because it is based on outdated methodologies, which do not have the capacity to provide novel insights into the role of variants on CFTR structure/function:

1. The use of expression minigenes (EMGs) and heterologous expression in HEK cells are outdated technologies with known artifacts. This type of study should have been done with CRISPR/Cas9 modification of the endogenous locus in airway and intestinal cell lines at minimum, since primary epithelial cells are rather difficult to engineer or, in iPSCs with subsequent differentiation in to relevant epithelial cells.

2. The authors have ignored an extensive literature on complex intronic and exonic elements that impact spicing, and on the fundamental role of chromatin features on splicing and RNA processing.

3. To define variants as "deep" intronic (eg Table 2) when they are 20 or 40 bp away from the splice site is erroneous. Many CFTR introns are greater than 10 kb long and it is known that there are well defined regulatory elements and splicing variants truly "deep" within these introns, several kb away from intron/exon boundaries. The variants described in this manuscript all lie within an extended splice site machinery.

4. How is the fragment analysis quantified in SFig 2, F2A, F3A, etc? It would seem difficult to reliably quantify PCR amplicons that are gel extracted prior to fragment analysis. The recovery efficiency of gel extraction is very low and variable from extraction to extraction. This could influence the quantification of transcripts derived from the same EMG (as presented in the mentioned figures).

5. If isoform 2 of c.2908 variants produces S912-G970 deletion (Figure 2A, left), and c.2908G>A produces more of this transcript than isoform 1, how can the authors be confident that the protein species detected for the EMG constructs of the two variants that cause missplicing are from isoform 1 and not isoform 2? The predicted protein sizes are not given, nor are any molecular weight markers noted on the western blot in Figure 2B. To better estimate the identity of this protein species, the authors should express the cDNA of these predicted protein species and compare to lanes 5 and 7 in the current western blot.

6. The conclusion made on lines 240-245 are not fully supported by the data presented. cDNA constructs were not made that would generate the predicted protein sequence of the misspliced c.2909 variants, and tested in short circuit current studies, therefore, it is only speculation that these immature protein species would be non-functional.

7. How does the short circuit current experiments for the stable lines in CFBE compare to the parental line (CFBE with Flp integration)? That is, how can the change in channel activity be attributed to the integrated cDNA construct and not to the effect of the drugs on the F508del proteins in these cells (CFBE/CFBE41o- is F508del homozygous), especially since Orkambi (lum/iva) can be prescribed for and ivacaftor has been reported to affect channel activity of F508del in cultured HBE cells (Van Goor, PNAS 2009, PMID: 19846789). The contribution of F508del to the signal is mentioned in the description of the results of figure 4C (pg 18), however again, it is unclear how the authors can conclude with the data presented, that the responses of the cells in figure 4C, especially in regards to ivacaftor alone, are contributed by the 3717+40G>A allele and not F508del.

8. Comprehensive raw data is not shown for “RNA effect” and “Protein present” columns for all variants in both tables, this should be provided in some form as supplementary data, and most especially for variants further explored in subsequent figures. It is unclear if protein production was assessed via western for each variant, or if predictions were made based on sequencing of RNA species. Additional information is also important in order to support statements like those made on line 330, in relation to the two intronic variants that express reduced levels of full-length transcript--was that calculated in relation to the WT EMG construct that harbored these mutations? How much reduction of transcript? Was it significantly different? A quantitative assay is needed to support this statement. This would also be helpful to support the conclusion that the “shortened protein isoform produced by this variant (from the EMG of c.454A>G)….” (line 487) is in fact producing a protein species.

9. HNEs with the genotype c2657+5G>A/W1282X are used to assess missplicing caused by the intronic variant, while comparing to a WT control (Figure 4A). The authors conclude that because W1282X is a nonsense variant, the presence of 9 reads that map from E15 to E16 and 3 reads that map from E15 to E17 as compared to the 27 reads in WT mapping from E15 to E16 mean that the intronic variant only causes partial missplicing. However, since W1282X is in exon 23, and is known to produce some detectable levels of mRNA transcript that can be measured under normal conditions (for example: Linde et al JCI, 2007, PMID: 17290305), the authors should use a W1282X homozygous HNE sample for control to fully conclude that the normal splicing detected between E15/E16 is not contributed by the W1282X allele.

Minor comments:

Please provide full details of antibodies that were used in this study.

It would be useful to remind readers that the CFBE/FLP cell line only has one copy of the FLP recombinase (reference 29), and that therefore integration copy number should not impact EMG or cDNA expression values between cell lines as all lines should have 1 integration

Please provide in the methods summary for “Collection and culture of primary HNEs” additional details on growth conditions (ie media used, disassociation reagent, concentration of ROCK inhibitor, approximately how long cells were cultured at ALI, etc)—having to sift through 4 references is tedious.

While the focus of this manuscript is on non-synonymous substitutions in exons, there have been multiple reports of synonymous substitutions that alter CFTR splicing. Reference of these studies could further strengthen the argument to further characterize variants by other means than just predicted protein sequence alterations.

How were the variants indicated in the tables identified? Were these all the variants from the CFTR Mutation Database (Toronto) or from the CFTR2 list? If the latter, what list version was used? (page 6, lines 127-129, page 15 lines 322-323)

What is the scale of the y-axis for the fragment analysis shown (SF1)?

Please provide the molecular weight markers for the western blot in Figure 2B, as well as numbering the lanes as described in the text on page 10.

References for previously reported primer sequences should be provided within the table

For HNE electrophysiology in Figure 4, the results note the % channel function as it corresponds to WT HNE (pg 18), however these traces/values are not provided.

**Have all data underlying the figures and results presented in the manuscript been provided?**

Reviewer #1: Yes

Reviewer #2: Yes

Reviewer #3: Yes

PLOS authors have the option to publish the peer review history of their article (what does this mean?). If published, this will include your full peer review and any attached files.

Reviewer #1: No

Reviewer #2: No

Reviewer #3: No

---

## [Editor Report · Decision Letter 1]

8 Sep 2020

Dear Dr Sharma,

We are pleased to inform you that your manuscript entitled "Evaluation of both exonic and intronic variants for effects on RNA splicing allows for accurate assessment of the effectiveness of precision therapies" has been editorially accepted for publication in PLOS Genetics. Congratulations!

Yours sincerely,

Gregory M. Cooper, PhD

Associate Editor

PLOS Genetics

Hua Tang

Section Editor: Natural Variation

PLOS Genetics

Comments from the reviewers (if applicable):

**Data Deposition**

http://datadryad.org/submit?journalID=pgenetics&manu=PGENETICS-D-20-00901R1

**Press Queries**

---

## [Editor Report · Acceptance letter]

14 Oct 2020

PGENETICS-D-20-00901R1 

Evaluation of both exonic and intronic variants for effects on RNA splicing allows for accurate assessment of the effectiveness of precision therapies 

Dear Dr Sharma, 

We are pleased to inform you that your manuscript entitled "Evaluation of both exonic and intronic variants for effects on RNA splicing allows for accurate assessment of the effectiveness of precision therapies" has been formally accepted for publication in PLOS Genetics! Your manuscript is now with our production department and you will be notified of the publication date in due course.

With kind regards,

Jason Norris

PLOS Genetics

On behalf of:
